# Fast and scalable production of crosslinked polyimide aerogel fibers for ultrathin thermoregulating clothes

Tiantian Xue[1], Chenyu Zhu[1], Dingyi Yu[1], Xu Zhang[2], Feili Lai[3], Longsheng Zhang[2], Chao Zhang ®[1], Wei Fan ®[1,2] ✉ & Tianxi Liu[1,2] ✉

Polyimide aerogel fibers hold promise for intelligent thermal management fabrics, but their scalable production faces challenges due to the sluggish gelation kinetics and the weak backbone strength. Herein, a strategy is developed for fast and scalable fabrication of crosslinked polyimide (CPI) aerogel fibers by wet-spinning and ambient pressure drying via UV-enhanced dynamic gelation strategy. This strategy enables fast sol-gel transition of photosensitive polyimide, resulting in a strongly-crosslinked gel skeleton that effectively maintains the fiber shape and porous nanostructure. Continuous production of CPI aerogel fibers (length of hundreds of meters) with high specific modulus (390.9 kN m kg$^{-1}$) can be achieved within 7 h, more efficiently than previous methods (>48 h). Moreover, the CPI aerogel fabric demonstrates almost the same thermal insulating performance as down, but is about 1/8 the thickness of down. The strategy opens a promisingly wide-space for fast and scalable fabrication of ultrathin fabrics for personal thermal management.

In recent years, intelligent thermoregulating textiles with personal thermal management capabilities have generated significant attention, which can minimize energy consumption through providing advanced thermal comfort for wearers under various environmental temperatures[1–3]. Specifically, for some occasions such as aerospace and fire scenes, lightweight and ultra-thin fabrics are needed to ensure the working efficiency and safety of the wearer[4–6]. The emerging aerogel fibers/fabrics, inheriting the three-dimensional (3D) porous structure of aerogels and the flexibility of fibers, show great potential in intelligent thermoregulating textiles due to their lightweight, high porosity and multi-integrated functions[7,8]. The highly porous structure can endow aerogel fibers with low thermal conductivity (23–50 mW m$^{-1}$ K$^{-1}$) that greatly suppresses heat loss. Besides, its unique microstructure can provide favorable conditions for the integration of smart materials, such as phase change materials[9]. Therefore, aerogel fibers and their composite fabrics demonstrate great potential in personal thermal management.

Currently, a few aerogel fibers with diverse functionalities such as silica, graphene, MXene, Kevlar, and polyimide (PI) aerogel fibers have been developed[7,9–14]. Compared to other organic or inorganic aerogel fibers, PI aerogel fibers have showed considerable potential in thermal management textiles in a wide range of temperature due to their good temperature resistance, and remarkable mechanical properties. However, insoluble PI are normally processed via the sol-gel transition of soluble poly(amic acid) (PAA) intermediate followed by thermal or chemical imidization[15–17]. This two-step reaction involves a complex and long-time sol-gel process of PAA, which cannot match with the short-time and continuous spinning process, giving rise to the difficulty in preparing PI aerogel fibers. To date, two main techniques have been reported to prepare PI aerogel fibers, i.e., (1) a sol-gel confined transition strategy, in which the sol-gel transition process is completed in the capillary tubes[16], and (2) a freeze-spinning technique based on water-soluble PAA salt[12,18]. However, the former cannot enable continuous preparation of aerogel fibers, while the latter results in aerogel

[1]State Key Laboratory for Modification of Chemical Fibers and Polymer Materials, College of Materials Science and Engineering, Donghua University, 2999 North Renmin Road, Shanghai 201620, China. [2]Key Laboratory of Synthetic and Biological Colloids, Ministry of Education, School of Chemical and Material Engineering, Jiangnan University, Wuxi 214122, P. R. China. [3] State Key Laboratory of Metal Matrix Composites, School of Materials Science and Engineering, Shanghai Jiao Tong University, Shanghai 200240, P. R. China. ✉e-mail: weifan@jiangnan.edu.cn; txliu@jiangnan.edu.cn

fibers with pore diameter up to tens of micrometers and relies on complicated equipment. Recently, PI aerogel fibers have been produced from organo-soluble PI via a wet-spinning approach combined with freeze-drying, avoiding the sol-gel process of PAA intermediate[19]. However, all the above strategy requires post-treatment after spinning process, such as supercritical-drying or freeze-drying, to maintain the highly porous structure of aerogel fibers, which is time- and cost-consuming. It still remains a great challenge for rapid preparation of PI aerogel fibers via ambient pressure drying.

Normally, aerogel fibers are fabricated through spinneret extrusion, sol-gel transition and supercritical/freeze drying processes. The sol-gel process is essential for the formation of typical three-dimensional porous aerogel structure. Two main obstacles exist in sol-gel transition and drying process respectively to produce a continuous aerogel fiber. The first one is the long-time sol-gel transition process. For the fabrication of aerogel fibers, the spinning solution should undergo a fast dynamic sol-gel transition after extrusion and maintain self-supporting fiber morphology, i.e., the gelation rate should match with the extrusion rate[4]. This is totally different from the fabrication of aerogel bulks, which undergoes static sol-gel transition in a mold, taking a long time ranging from several hours or even days[20,21]. The second obstacle lies in the drying process, in which the porous structure is easy to collapse due to the weak skeletal strength, particularly for ambient pressure drying[22]. The high-throughput fabrication of high-performance aerogel fibers faces the following challenges: (1) achieving fast dynamic sol-gel transition of the spinning solution, and (2) constructing a high-strength gel skeleton and effectively avoiding skeleton collapse during ambient pressure drying.

Herein, we report a fast and scalable fabrication strategy for crosslinked polyimide (CPI) aerogel fibers by wet-spinning and cost-effective ambient pressure drying via UV-enhanced dynamic gelation strategy. A spinning solution based on photosensitive polyimide (PPI) with trifluoromethyl group is designed and synthesized. The UV-enhanced dynamic gelation can induce rapid crosslinking of PPI spinning solution under UV light within 10 s and enable fast sol-gel transition, resulting in a strongly-crosslinked gel skeleton that can effectively prevent the structural collapse of CPI aerogel fibers during the ambient pressure drying process. Fast and continuous fabrication of high strength aerogel fibers with length of hundreds of meters are achieved within 7 h using our strategy, which is appreciably faster than the previously-reported strategy (normally 49–94 h). The ultrathin aerogel fabric with thickness of 0.7 mm, roughly 1/8 of that of down, exhibits almost the same thermal insulating performance as down. Furthermore, for the proof-of-concept study, the CPI aerogel fabrics can be integrated with shape memory materials to build intelligent thermally adaptive fabrics for personal thermal management. Our research offers a scalable production method for developing aerogel fibers and textiles to meet advanced application scenarios.

## Results
### Preparation and morphology of CPI aerogel fibers
The crosslinked polyimide (CPI) aerogel fiber has been prepared by wet-spinning, solvent exchange and ambient pressure drying via UV-enhanced dynamic gelation strategy (Fig. 1a & Supplementary Movie 1). First, organo-soluble PI was synthesized through one-pot copolymerization of dianhydride (4,4′-hexafluoroisopropylidene di(phthalic anhydride), 6FDA) with two types of diamines (4,4′-diaminodiphenyl ether, ODA and 3,5-diaminobenzoic acid, DABA) followed by imidization at elevated temperatures (Step 1 and 2 in Supplementary Fig. 1). Fourier-transform infrared (FTIR) and nuclear magnetic resonance (NMR) spectra evidence that the organo-soluble PI with

carboxyl groups is successfully prepared by the one-pot copolymerization. The disappearance of the absorption peak of amide bond at 1608 cm⁻¹ and 1546 cm⁻¹ and appearance of new peaks (C–N–C, 1355 cm⁻¹) in the FTIR spectrum of PI indicates that the poly(amic acid) (PAA) is completely imidized during the heating process[23] (Supplementary Fig. 2a, b). The complete conversion of PAA to the PI was also confirmed by the disappearance of the amide (-NHCO-) at 10.5–11.0 ppm in the ¹H NMR spectrum of PI[24] (Supplementary Fig. 2c). The chemical structure of PI was further confirmed by ¹H NMR, ¹³C NMR, ¹⁹F NMR, and FTIR spectra (Supplementary Fig. 2). The synthesized PI is soluble in organic solvents such as N-methylpyrrolidone (NMP), N,N-dimethylformamide (DMF) and N,N-dimethylacetamide (DMAc) and so on (Supplementary Table 1), since the introduction of the -CF₃ group in the PI main chain can effectively increase the free volume between the molecular chains, reduce the regularity of the molecular chain segments, and reduce the π-π interactions between the benzene rings, resulting in the good solubility[25]. The resulting organo-soluble PI showed high molecular weight ($M_w$ - 30273 g mol⁻¹) and narrow polydispersity index (PDI - 1.63) (Supplementary Fig. 3). Then, the photosensitive polyimide (PPI) was synthesized by grafting β-hydroxyethyl methacrylate (HEMA) onto the organo-soluble PI chain via Steglich esterification between the carboxyl groups on PI and hydroxyl groups on HEMA (Step 3 in Supplementary Fig. 1). The PPI-x with different grafting ratios of HEMA was successfully prepared (where x represents grafting ratio, Supplementary Table 2), as demonstrated by the FTIR and NMR spectra (Supplementary Fig. 4a–c). According to the FTIR spectrum, compared with pure PI, additional peaks of -CH₃ bonds at 2924 cm⁻¹ and -CH₂ bonds at 2860 cm⁻¹ appear for PPI-x samples, suggesting effective grafting of HEMA onto the PI[26,27]. The aliphatic hydrocarbon chains of HEMA in the PPI are indicated by the peaks at about 6.06 ppm, 4.6 ppm, and 4.45 ppm in the ¹H NMR spectra, which correspond to $H_{21}$, $H_{22}$, and $H_{24}$[28], respectively. The practical grafting ratio calculated by ¹H NMR are 24.8%, 47.9% and 76.0% for PPI-25, PPI-50 and PPI-100, respectively, which are comparable with the theoretical grafting ratio (Supplementary Fig. 4d). The molecular weight of PPI is measured as 31205 g mol⁻¹, which is applicable to typical photo-crosslinking processes[29] (Supplementary Fig. 5).

The PPI exhibits excellent solubility and stability in common solvents, which ensures the formation of spinning solution and improves processability (Supplementary Fig. 6 & Supplementary Table 3). Subsequently, the spinning solution was prepared by dissolving PPI and photoinitiator (Irgacure2100) in NMP, which showed high zero-shear viscosity and shear-thinning behavior, benefiting for the spinning process (Supplementary Fig. 7). The spinning solution was extruded to form a filament, which can be rapidly transformed into a gel fiber under the irradiation of UV light. After subsequent solvent exchange and ambient pressure drying, the CPI aerogel fibers were finally obtained. In the UV-enhanced dynamic gelation strategy, the randomly distributed PPI chains are rapidly crosslinked by UV irradiation, resulting in a highly crosslinked and high-strength gel network (Fig. 1b). Moreover, the rapid gelation process and ambient pressure drying strategy provide great flexibility for fast, continuous and scalable fabrication of CPI aerogel fibers. The resulting CPI aerogel fibers clearly show a perfect fiber morphology with a diameter of ~300 μm demonstrated by scanning electron microscopy (SEM) image in Fig. 1c. The corresponding high-magnification SEM image shows a highly porous nanostructure formed by 3D interconnected nanofiber network, demonstrating a typical aerogel morphology[30,31] (Fig. 1d). Figure 1e shows a roll of as-prepared CPI aerogel fibers with length of hundreds of meters, which can be woven into large-size CPI aerogel fabric with length larger than 1 meter by semi-automatic weaving machine (Fig. 1f). Therefore, our strategy is expected to be a promising alternative for the high-throughput and scalable fabrication of aerogel fibers/fabrics.

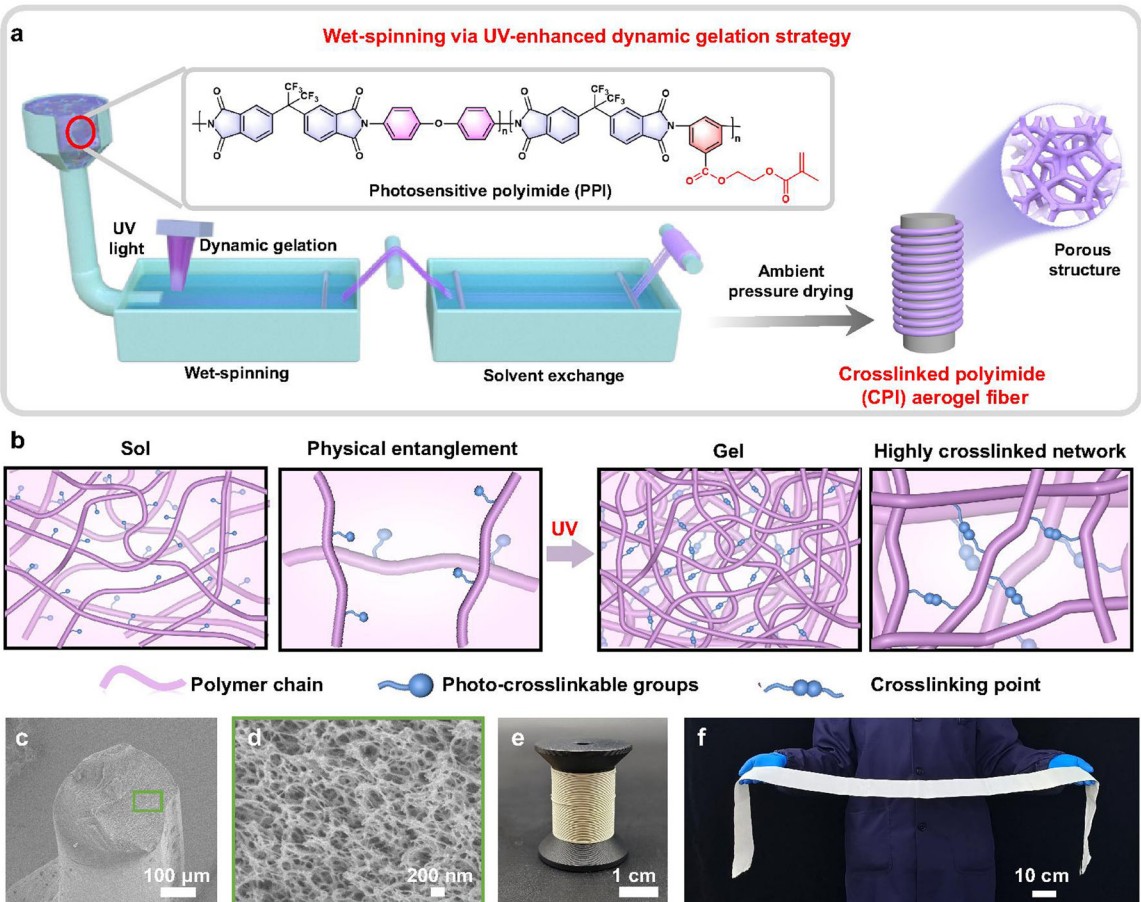

**Fig. 1 | Preparation and morphology of CPI aerogel fibers. a** High-throughput fabrication of aerogel fibers by wet-spinning, solvent exchange and ambient pressure drying via UV-enhanced dynamic gelation strategy. **b** Illustration of sol-gel transition processes by UV irradiation. Cross-sectional SEM image at (**c**) low and (**d**) high magnifications. **e** Photograph of CPI aerogel fibers and (**f**) corresponding woven fabric (0.1 m × 1.2 m).

## Sol-gel transition process of PPI via UV-enhanced dynamic gelation strategy

Under the irradiation of UV light, the double bonds of PPI are induced to free radical polymerization to form a strong crosslinked polyimide gel (Supplementary Fig. 8). The characteristic vinyl peak (C = C stretching) of PPI-100 at 1628 cm⁻¹ gradually disappears with the increase of UV irradiation time as shown in FTIR spectra in Fig. 2a. A high double-bond conversion rate of 93.2% is achieved under 30 s UV irradiation, according to calculations based on variations in the absorption peak area of the double bond (Fig. 2b)[32]. For UV-enhanced dynamic gelation strategy, the gelation rate is close to the reaction rate ($R_p$) of free radical polymerization.

$$R_p = -\frac{d[M]}{dt} = k_p[M]\left(\frac{R_i}{2k_t}\right)^{1/2} \quad (1)$$

$$[M] = (1 - X) * [M]_0 \quad (2)$$

$$R_i = 2fk_d[S] \quad (3)$$

where $k_p$ is the propagation rate constant, $[M]$ is the monomer concentration at the reaction time of $t$, $[M]_0$ is the monomer concentration before the reaction, $X$ is the conversion rate of double bond at the reaction time of $t$, $k_t$ is the chain termination rate constant, $R_i$ is the photoinitiation rate, which is related to the photoinitiation efficiency

($f$), decomposition rate constant of photoinitiation ($k_d$) and initiator concentration ([$S$]).

Therefore, the above formula (1) could be converted into:

$$\ln\frac{1}{[1-X]} = k_p\left(\frac{fk_d}{k_t}\right)^{1/2}[s]^{1/2}t \quad (4)$$

According to Eq. (4), the gelation rate constant of UV-enhanced dynamic gelation strategy obtained by linear fitting is $10.2 \times 10^{-2}$ L mol⁻¹ s⁻¹, as shown in the quantitative results in Fig. 2c. The UV-enhanced dynamic gelation strategy shows much faster gelation rate constant than previous reported work ($4.8 \times 10^{-2}$ L mol⁻¹ s⁻¹)[33]. Besides, the presence of the methylene carbon at 44 ppm in the ¹³C NMR spectrum of CPI also indicates that the crosslinked structure is successfully formed (Supplementary Fig. 9). The structure evolution during the sol-gel transition process of PPI was simulated by coarse-grained molecular dynamics simulation (Fig. 2d & Supplementary Fig. 10). For the PPI solution, the molecular chains are gradually crosslinked by cross-linkable functional groups to form a continuous gel network. Moreover, in the UV-enhanced dynamic gelation strategy, the gelation time is critical for the high-throughput preparation of aerogel fibers. Figure 2e shows the fast gel formation for the PPI-100 solution, which loses its fluidity and transforms into a gel after 10 s of UV irradiation. To further quantify the gelation process, a dynamic time-scan rheology experiments were performed with in-situ photo-rheometer[34]. As illustrated in Fig. 2f, the curves of storage modulus and loss modulus rapidly cross over at 10 s of UV

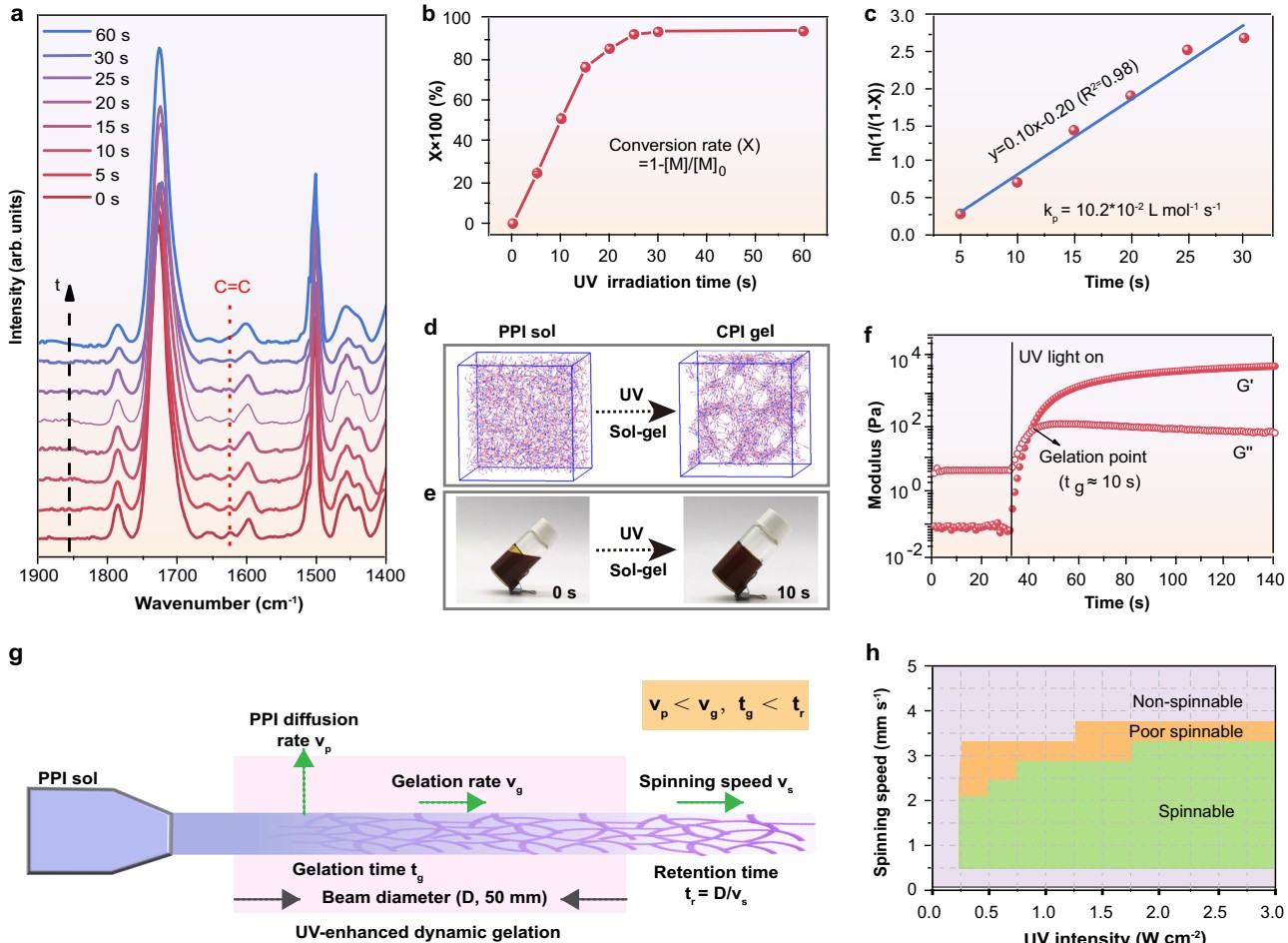

**Fig. 2 | Sol-gel transition process of PPI via UV-enhanced dynamic gelation strategy. a** FTIR spectra of PPI-100 at different times of UV irradiation. **b** The double bond conversion rate of PPI-100 at different UV irradiation time. **c** Linear regression of conversion rate (X) versus UV irradiation time (t). $k_p$ is the propagation rate constant. **d** Coarse-grained molecular dynamics simulation of sol-gel transition of PPI solution. **e** Photographs showing gelation process of PPI-100 solution under UV irradiation with 10 s. **f** Moduli versus time from photo-rheology measurements of the PPI-100 solution. UV irradiation started at 30 s. **g** Schematic illustration of spinning kinetics. **h** Phase diagram showing the spinnability of PPI-100 spinning solution. Colors in the graphs are: purple (Non-spinnable), yellow (Poor spinnable) and green (Spinnable).

irradiation, corresponding to the gelation point of PPI-100. In addition, the gelation time of PPI solution increases with the decrease of grafting ratio of HEMA (Supplementary Fig. 11), with the fastest gelation of 10 s for PPI-100.

The spinning kinetics of wet spinning via UV-enhanced dynamic gelation strategy is shown in Fig. 2g. In the spinning process, parameters including the diffusion rate ($v_p$) of the PPI in the coagulation bath, the gelation rate/time ($v_g/t_g$), the spinning speed ($v_s$) and the retention time ($t_r$, ratio of beam diameter D to $v_s$) combine to influence the formability and structure of the CPI gel fibers. Importantly, since the sol-gel transition occurs under dynamic movement during spinning process, it is critical to achieve a match between $v_s$ and $v_g$ for a good formability of the aerogel fibers. Due to the fast gelation rate constant ($10.2 \times 10^{-2}$ L mol$^{-1}$ s$^{-1}$) of UV-enhanced dynamic gelation strategy, the diffusion rate is much smaller than the gelation rate (i.e., $v_p < v_g$). This can effectively maintain the fiber shape after extrusion of PPI solution, avoiding its diffusion caused by the concentration difference between the coagulation bath and the spinning solution. For the ultimate formation of gel fiber, the $t_g$ and $t_r$ are equally crucial. The gelation time can be shortened when UV intensity increases, where the $t_g$ in the range of 5–20 s is defined as the spinnable region under various UV intensities (Supplementary

Fig. 12). When the $t_g$ is less than 5 s, the needle tends to be clogged due to the rapid gelation process under UV irradiation. The extruded PPI, however, is unable to retain the self-supporting fiber morphology when the $t_g$ extends to over 20 s. The spinnability phase diagram shows that PPI is spinnable at spinning speeds of 0.4 to 3.3 mm s$^{-1}$ (Fig. 2h). Therefore, the retention time ($t_r = D/v_s$) is in the range of 15–125 s, much greater than the gelation time (10 s) of PPI-100 under UV intensity of 0.6 W cm$^{-2}$, which ensures a complete gelation reaction and the formation of stable gel fibers. However, the PPI fails to complete gelation when the spinning speed exceeds 3.3 mm s$^{-1}$, and discontinuous fibers are produced. Additionally, the gelation depth of PPI-100 is calculated to be 2.3 mm according to the Jacob's equation and linear fit (Supplementary Fig. 13a), which is significantly greater than the fiber diameter (<1 mm), ensuring a complete gelation in the radial direction of the fibers and forming a homogeneous gel network. With increasing exposure time under UV intensity of 0.6 W cm$^{-2}$, the gelation depth of PPI-100 increases allowing for the creation of aerogel fibers with various diameters (Supplementary Fig. 13b, c). Therefore, it is apparent that the UV-enhanced dynamic gelation strategy exhibits ultra-fast gelation and high controllability, which are essential to ensure high-throughput production of aerogel fibers.

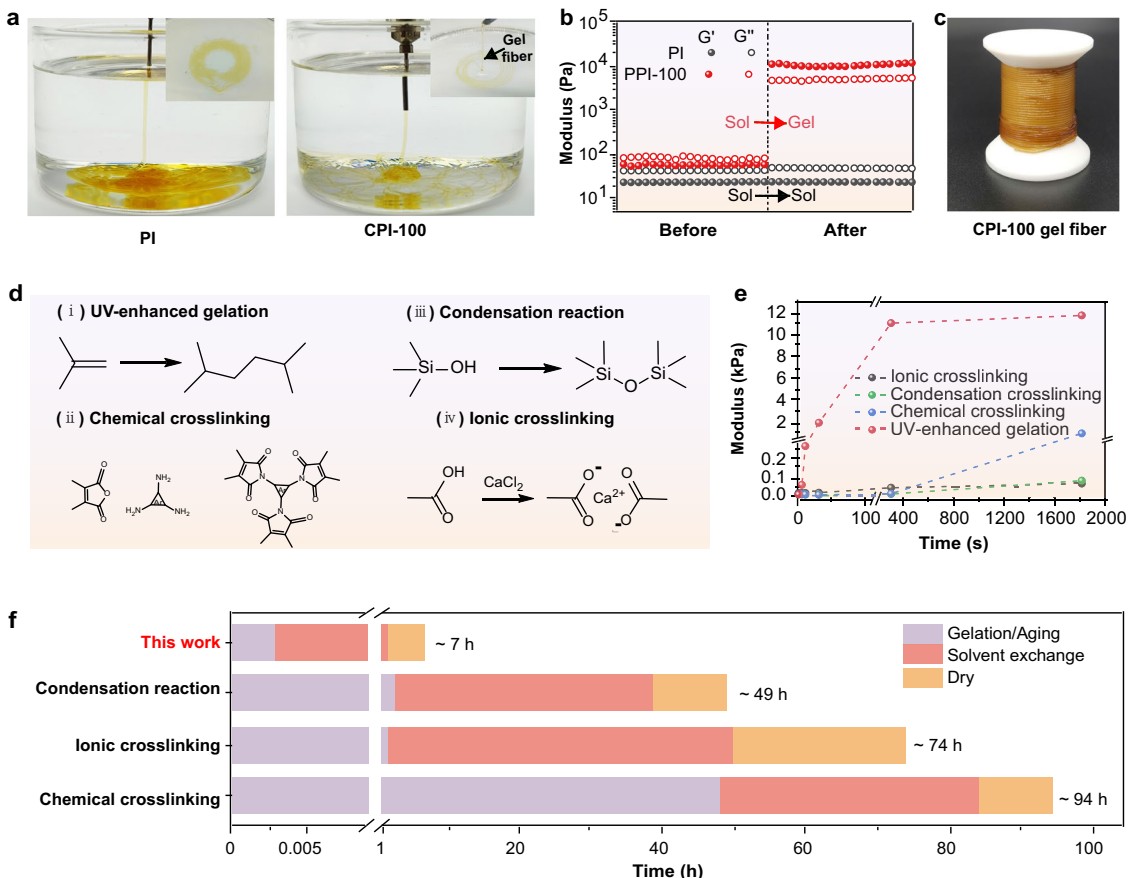

**Fig. 3 | Gelation and backbone strength of CPI gel fibers. a** Photographs showing the extrusion of PI (left) and PPI-100 (right) spinning solution into the NMP solvent under UV radiation. Inset shows the spun fibers collected in coagulating bath. **b** Storage modulus (G′) and loss modulus (G″) of PI (gray) and PPI (red) solution before and after UV irradiation. **c** Photograph of the CPI gel fibers. **d** Schematic illustration of the chemical structure evolution, and (**e**) gelation kinetics of UV-enhanced gelation strategy (red), condensation reaction strategy (green), chemical crosslinking strategy (blue) and ionic crosslinking strategy (gray), respectively. **f** Time consumption of preparing aerogel fibers through our strategy compared to previous gelation strategies. Colors in the graphs are: purple (Gelation/Aging time), salmon (solvent exchange) and yellow (Drying time).

## Gelation and backbone strength of CPI gel fibers

The sol-gel transition process is further visually demonstrated by extruding PI and PPI spinning solution into the NMP solvent under UV radiation (Fig. 3a). After extrusion, the PI diffuses in the solvent quickly and cannot maintain the fiber shape (Supplementary Movie 2), while the PPI-100 can form stable crosslinked gel fibers in the solvent, indicating an effective UV-induced sol-gel transition (Supplementary Movie 3). The change of storage modulus and loss modulus of PI and PPI solution before and after UV irradiation is shown in Fig. 3b. The storage modulus and loss modulus of PPI-100 solution increase by an order of magnitude after UV irradiation. In addition, the storage modulus (~11,700 Pa) of PPI-100 is much higher than loss modulus (~5020 Pa) after UV irradiation, presenting a gel status[35]. However, the modulus of the PI solution after UV irradiation does not change significantly and the storage modulus is smaller than the loss modulus, indicating a sol status. More importantly, CPI gel fibers prepared by UV-enhanced dynamic gelation strategy exhibit high tensile strength and elastic modulus, and the strength increases with the grafting ratio of HEMA (Supplementary Fig. 14a). This is attributed to that the PPI with high grafting ratio can provide more crosslinking sites for the gel formation, thus endowing the resultant gel fibers with a high crosslinking density (Supplementary Fig. 14b). As a result, the gel fibers can be further collected into a roll without shape damage due to the stable and high-strength gel backbone (Fig. 3c), which can effectively inhibit shrinkage during solvent exchange (Supplementary Fig. 15) and

facilitate the construction of 3D porous structures after drying. To further demonstrate the advantages of UV-enhanced dynamic gelation strategy in constructing aerogel fibers, the gelation kinetics are compared with the previously reported gelation strategies (e.g., condensation reaction[10,36,37], chemical crosslinking[16], and ionic crosslinking[11]) (Fig. 3d). We recorded the variation of the modulus of the gels with time obtained by several gelation strategies (Fig. 3e). The UV-enhanced gelation strategy induces a fast sol-gel transition in the precursor solution while conferring high modulus of gels (~11,700 Pa). In contrast, the modulus of gels produced by condensation reactions, chemical crosslinking and ionic crosslinking is less than 10 Pa. Such weak gel skeletons inevitably need to be supercritical- or freeze-dried to remove the solvent in order to prevent collapsing of 3D skeleton. In addition, these gelation strategies exhibit sluggish gelation kinetics, which is caused by the low diffusion rate of the crosslinking or gelling agent (such as $Ca^{2+}$, urea) during the spinning process. Significantly, the high-strength gel backbone prepared by UV-enhanced dynamic gelation strategy with rapid gelation kinetics prevents structural collapse of the aerogel fiber during ambient pressure drying. Moreover, the CPI-100 with abundant methyl/trifluoromethyl groups exhibits hydrophobic surface and low capillary pressure (Supplementary Fig. 16), which can further avoid structural collapse during ambient pressure drying. As a result, it only takes about 7 h to prepare aerogel fibers using our strategy via wet spinning, solvent exchange, and ambient pressure drying (Fig. 3f). Furthermore, the entire process of

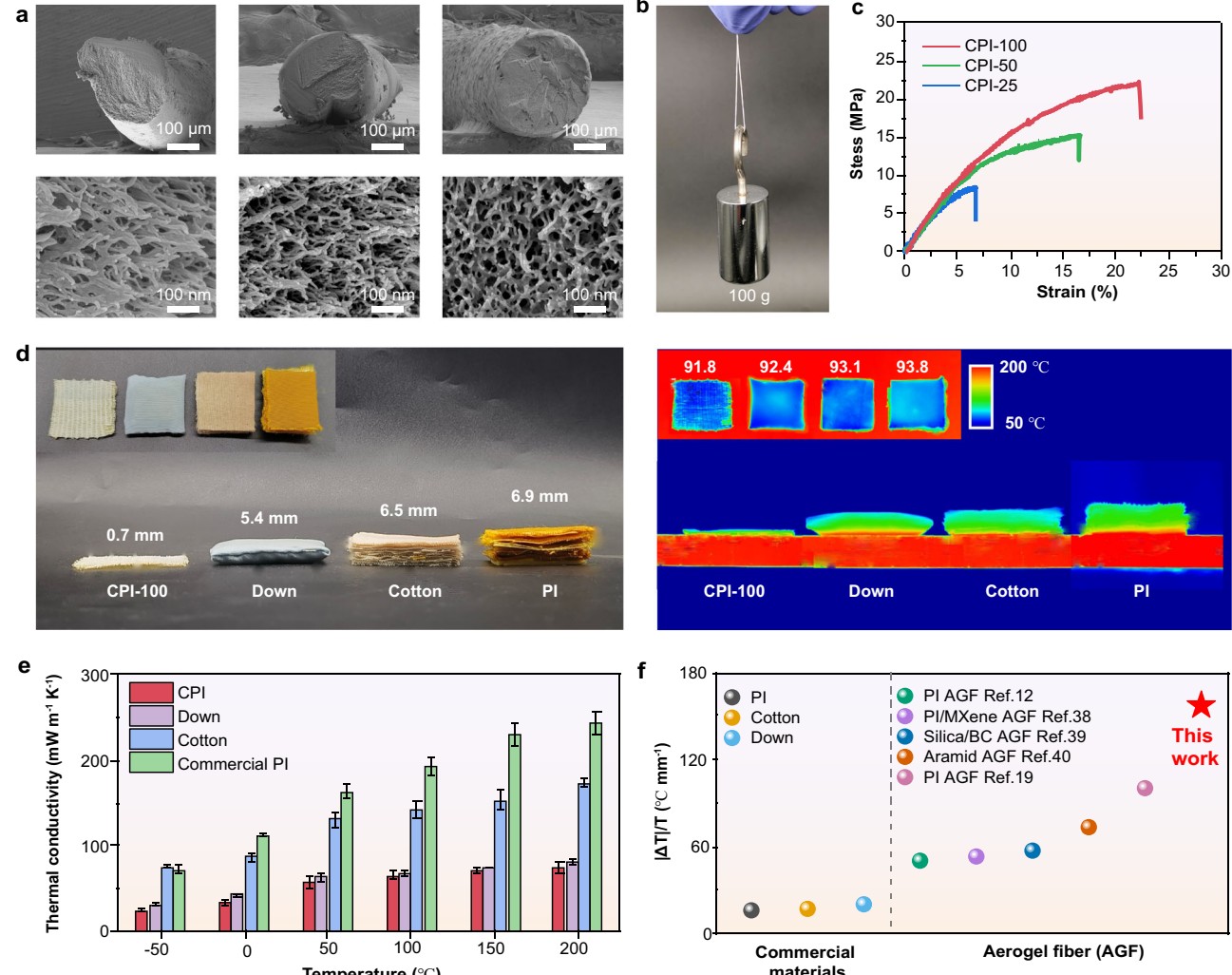

**Fig. 4 | Morphology and physical performance of CPI fibers/fabrics. a** SEM images of CPI aerogel fibers fabricated from PPI with different grafting ratios. **b** A single CPI-100 aerogel fiber can load a weight of 100 grams. **c** Tensile stress-strain curves of CPI-25 (blue), CPI-50 (green) and CPI-100 (red) aerogel fibers. **d** Optical and infrared thermal images of CPI-100 aerogel fabric (thickness = 0.7 mm), down (thickness = 5.4 mm), cotton (thickness = 6.5 mm) and PI (thickness = 6.9 mm) placed on a hot stage of 200 °C. **e** Thermal conductivity of the CPI aerogel fabric, down, commercial PI and cotton at −50-200 °C. Bars represent the mean, and error bars represent the standard deviation. Colors in the graphs are: red (CPI), purple (Down), blue (Cotton) and green (Commercial PI). **f** Summary of temperature difference/thickness (|ΔT |/T) values for commercial materials and reported aerogel fibers. |ΔT| means temperature difference between upper surface temperature of materials and hot stage. Colors in the graphs are: gray (PI), yellow (cotton), blue (Cotton), green (Down), purple (PI/MXene AGF), navy (Silica/Bacterial Cellulose AGF, Silica/BC AGF), brown (Aramid AGF), pink (PI AGF) and red (This work, CPI aerogel fabric).

producing aerogel fibers is continuous and scalable. In contrast, the time consumption of the fabrication of aerogel fibers reported previously was up to 94 h, which is caused by the slow gelation kinetics (Supplementary Table 4)[10,11,16,36,37]. Besides, due to the weak gel skeleton, gel fibers can only be prepared by energy-intensive supercritical-drying or freeze-drying, while continuous preparation of aerogel fibers is unachievable. As demonstrated above, our strategy exhibits obvious advantages over the previous ones for manufacturing aerogel fibers with high throughput and low energy consumption.

**Morphology and physical performance of CPI fibers/fabrics**
The microstructure of CPI aerogel fibers fabricated from PPI with different grafting ratios was observed by SEM (Fig. 4a). As shown in the SEM images, CPI-25, CPI-50 and CPI-100 all displays typical fiber morphology and 3D interconnected nanofiber porous structure, as previously reported for polyimide aerogels[13]. However, after removal of the UV dynamic gelation process, the PPI polymer chains started to aggregate together in the presence of non-solvents, resulting in the PPI

fibers displaying a typical finger-like pore structure (Supplementary Fig. 17). In particular, CPI-100 exhibits fiber morphology with low shrinkage and regular circle cross section, while CPI-25 and CPI-50 exhibit irregular cross section. In the high-magnification SEM images, CPI-100 exhibits relatively loose 3D interconnected nanofiber networks with pore size distribution around 50-250 nm (Supplementary Fig. 18). This is attributed to the highly crosslinked strong network of CPI-100 that avoids structural collapse during ambient pressure drying, resulting in stable and robust CPI aerogel fibers, which can load a weight of 100 grams (Fig. 4b). The resultant CPI-100 aerogel fibers exhibit low shrinkage (17.9%), low density (0.55 g cm$^{-3}$) and high specific modulus (390.9 kN m kg$^{-1}$) (Fig. 4c & Supplementary Fig. 19). Compared with CPI-25, CPI-100 aerogel fibers show an increase of tensile strength from 6.5 MPa to 22 MPa and modulus from 80 MPa to 215 MPa (Supplementary Fig. 20). Benefiting from the high mechanical properties of CPI-100 aerogel fibers, they can be further woven into aerogel fabrics, which can withstand a weight of 500 grams without breaking (Supplementary Fig. 21a, b). In addition, CPI-100 aerogel

fabrics have flexibility and structural stability in bending and folding tests, which makes them appropriate for wearable applications (Supplementary Fig. 21c, d).

The corresponding woven CPI-100 aerogel fabrics exhibit remarkable thermal insulating properties compared to certain commercial materials such as down, cotton, and PI fabric. When being placed on a hot stage with 200 °C, CPI aerogel fabric with thickness of 0.7 mm shows a temperature difference of 108 °C, comparable to that of down with 5.4 mm thick, cotton fabric with 6.5 mm thick and PI fabric with 6.9 mm thick (Fig. 4d). Therefore, the CPI aerogel fabric exhibits almost the same thermal insulating performance as down, while the thickness is roughly 1/8 of that of down. Besides, the CPI aerogel fabric also shows good thermal insulating properties on stages with different temperatures from −10 °C to 200 °C (Supplementary Fig. 22). Quantitatively, the CPI aerogel fabric shows a low thermal conductivity ($\lambda$) of 24.2 mW m$^{-1}$ K$^{-1}$ at −50 °C, and still remains a low $\lambda$ of 70.2 mW m$^{-1}$ K$^{-1}$ when the temperature increases to 150 °C (Fig. 4e). In contrast, the $\lambda$ of cotton fabric increases from 75.2 mW m$^{-1}$ K$^{-1}$ to 153.2 mW m$^{-1}$ K$^{-1}$ and that of commercial PI increases from 82.4 mW m$^{-1}$ K$^{-1}$ to 230.1 mW m$^{-1}$ K$^{-1}$ as temperature rises from −50 °C to 150 °C. The low thermal conductivity of the CPI aerogel fabric is attributed to its nanopore size that can effectively extend the solid heat conduction pathway and inhibit gas heat conduction. Therefore, the CPI aerogel fabrics can be potentially applied for ultrathin thermal insulating clothing. When a thermal puppet (58.0 °C) wears CPI-100 aerogel fabric, the temperature of puppet remains at 56.4 °C after 300 s, while that of commercial cotton fabric drops rapidly to 49.0 °C, indicating a good heat preservation performance of CPI-100 aerogel fabric (Supplementary Fig. 23). Compared with commercial materials and reported aerogel fibers[12,19,38–40], the CPI aerogel fiber shows excellent thermal insulating performance with ultrathin thickness (Fig. 4f & Supplementary Table 5).

**Intelligent thermoregulating performance of intelligent thermally adaptive textile**

Due to their interconnected porous network and high porosity, CPI aerogel fibers are favorable for the integration of smart materials, such as phase change materials (PCM), demonstrating great potential in intelligent thermoregulating textiles[41]. As a proof of concept, an intelligent thermally adaptive (ITA) textile is designed by integrating the shape memory CPI/PCM composite fabrics as interlayers for fire-fighting clothing (Supplementary Fig. 24). In detail, shape memory CPI/PCM fabrics were fabricated by impregnating paraffin wax, a typical PCM, into CPI aerogel fibers (Supplementary Fig. 25a). FTIR spectra and X-ray diffraction (XRD) patterns confirmed the successful composition of PCM with CPI aerogel fiber (Supplementary Fig. 25b, c). The CPI/PCM composite fiber shows a high loading of PCM (up to 80% according to thermogravimetric analysis (TGA) curves in Supplementary Fig. 25d) and phase change temperature of 65–70 °C according to differential scanning calorimetric (DSC) curve in Supplementary Fig. 25e. Besides, PCM filling in the porous structure of CPI aerogel fibers can improve the mechanical strength of PI/PCM composite fibers (Supplementary Fig. 25f). The CPI/PCM fiber/fabric can be programmed above the phase change temperature and fixed at room temperature to obtain a temporary shape, and return to the initial shape after re-heating (Supplementary Figs. 26, 27). Then, two pieces of CPI/PCM fabrics were inserted vertically between two pieces of commercial PI fabrics, forming an air gap between commercial PI fabrics (permanent shape in Supplementary Fig. 28a). The integrated ITA textile can be programmed above the phase change temperature and then cooled to room temperature to fix the temporary shape (bottom in Supplementary Fig. 28a). When the environment temperature is higher than the phase change temperature, the shape memory CPI/PCM fabric recovers from temporary shape to permanent shape and the ITA textile begins to expand to prevent heat transfer

(Supplementary Movie 4). Photographic and infrared thermal images in Supplementary Fig. 28b show the shape memory recovery behavior of ITA textile. When the temporary shape is placed on the hot stage (100 °C), the ITA textile begins to expand and gradually returns to its initial permanent shape. At the same time, the surface temperature of ITA textile decreases from 55.2 °C to 42.1 °C, demonstrating intelligent thermal adaptivity. The shape fixity ($R_f$) and shape recovery ($R_r$) of the ITA textile are 77.4% and 89.8%, respectively (Supplementary Fig. 29). More importantly, this ITA textile can be recycled and remain almost the same thermally adaptive behavior after secondary programming (Supplementary Fig. 30). This is mainly attributed to the stable thermal behavior of CPI/PCM fabrics in heating-cooling cycles (Supplementary Fig. 31). Therefore, the ITA textile with extraordinary deformation capabilities shows great potential in personal thermal management for military and industrial applications.

## Discussion

In summary, crosslinked polyimide aerogel fibers have been facilely prepared by wet-spinning via UV-enhanced dynamic gelation strategy. The photosensitive polyimide can crosslink and gelling under UV irradiation within 10 s, enabling the sol-gel transition immediately and maintaining fiber shape after extrusion. The strongly-crosslinked gel skeleton of CPI exhibits a high storage modulus of ~11,700 Pa, and the structural collapse of the aerogel fiber can thus be effectively prevented during the solvent exchange and ambient pressure drying process. The obtained CPI-100 aerogel fibers show high specific modulus (390.9 kN m$^{-1}$ kg$^{-1}$) and can be woven into aerogel fabrics. The ultrathin aerogel fabric (0.7 mm thick) exhibits impressive thermal insulating performance, showing a temperature difference of 108 °C when placed on a 200 °C hot stage, which is comparable to a 5.4 mm thick down jacket. Moreover, for the proof-of-concept study, an intelligent thermally adaptive textile is designed by integrating CPI aerogel fibers with phase change materials, achieving intelligent thermal regulation in hot environments. This work provides a broad possibility for the scalable and cost-effective fabrication of high-performance and multifunctional aerogel fibers, which demonstrates great potential for personal thermal management and beyond.

## Methods
### Materials
4,4′,-diaminodiphenyl ether (ODA, 98.0%), 3,5-diaminobenzoic acid (DABA, 98.0%), 4,4′-hexafluoroisopropylidene di(phthalic anhydride) (6FDA, 99%), 3,3′,4,4′-biphenyl tetracarboxylic dianhydride (BPDA, 98.0%), 1,3,5-tris(4-aminophenoxy) benzene (TAB) and 2,2′-dimethyl-benzidine hydrochloride were purchased from TCI Co, Ltd. β-hydroxyethyl methacrylate (HEMA, 99%), 4-dimethylaminopyridine (DMAP, 98%), CaCl$_2$, acetic acid, hexadecyltrimethylammonium bromide (CTAB, 99%), methyltrimethoxysilane and dicyclohexyl carbodiimide (DCC, 98%), pyridine, isoquinoline and photo-initiator (Irgacure 2100) were acquired from Aladdin Chemistry Co., Ltd, China. *N*,*N*-dimethylformamide (DMF, 99%), *N*,*N*-dimethylacetamide (DMAc, 99%), dimethyl sulfoxide (DMSO), dichloromethane (CH$_2$Cl$_2$), tri-chloromethane (CHCl$_3$), tetrahydrofuran (THF), *N*-methylpyrrolidone (NMP, 99%), acetone, paraffin wax, acetic anhydride and alcohol were bought from Shanghai energy chemical Co. Ltd., China.

**Preparation of organo-soluble polyimide (PI)**
The organo-soluble PI was synthesized by copolymerization and imidization[42]. The molar ratio of dianhydride (6FDA) and diamine (ODA and DABA) was set as 1.03:1, while that of DABA and ODA was 1:1. In short, DABA (10 mmol, 1.492 g), ODA (10 mmol, 2.002 g) and NMP (28 mL) were mixed and stirred under nitrogen until the diamines were completely dissolved. Then 6FDA (20.6 mmol, 9.1511 g) was added into the solution with continuous mechanical stirring to obtain poly(amic acid) (PAA). Then, 0.2 g of isoquinoline was added to the above

solution. Finally, the temperature was increased to 120 °C (1 h), 160 °C (1 h) and 200 °C (10 h) respectively to accomplish the imidization. The yield of PI is as high as 98.1%. $^1$H-NMR (DMSO-$d_6$, 600 MHz), $\delta_H$ (ppm): 13.48 (s, H, $H_{21}$), 8.12-8.25 (m, 4H, $H_6$), 7.96 (s, 5H, $H_7$ and $H_{16}$), 7.85 (s, 4H, $H_3$), 7.73–7.82 (d, 2H, $H_{17}$), 7.45–7.54 (d, 4H, $H_{12}$) 7.2–7.29 (d, 4H, $H_{13}$). $^{13}$C NMR (600 MHz, DMSO-$d_6$), $\delta_C$ (ppm): 166.6 ($C_{20}$), 166.3 ($C_{10}$), 166.1 ($C_1$), 156.5 ($C_4$, $C_{14}$), 137.8 ($C_{15}$), 136.4 ($C_2$), 133.5 ($C_7$), 131.2 ($C_{18}$), 130.5 ($C_5$), 129.8 ($C_6$), 128.3 ($C_{12}$), 127.6 ($C_3$), 126.8 ($C_{11}$), 124.9 ($C_{13}$), 124.0 ($C_{17}$), 122.9 ($C_{16}$), 121.0 ($C_9$), 64.9 ($C_8$). $^{19}$F NMR (DMSO-$d_6$, 600 MHz), $\delta_F$ (ppm): −63.63 (-$CF_3$). FTIR (ATR): 1786 cm$^{-1}$ (C = O stretching), 1717 cm$^{-1}$ (C = O stretching), 1500 cm$^{-1}$ (C = C), 1356 cm$^{-1}$ (C-N-C stretching), 1240 cm$^{-1}$ (C-F), 1099 cm$^{-1}$ (Ar-C-O symmetric stretching).

### Preparation of photosensitive polyimide (PPI)

The fabrication of photosensitive polyimide was achieved by grafting acrylate onto the above prepared polyimide. In detail, HEMA (10.2 mmol, 1.326 g), DCC (10.2 mmol, 2.104 g) and DMAP (1.02 mmol, 0.124 g) were sequentially added to the above PI solution for Steglich esterification (25 °C, 24 h). Then, the resulting solution was centrifuged and precipitated into ethanol to obtain the PPI-100. The PPI prepared with different grafting ratios of HEMA (25%, 50% and 100%) were designated as PPI−25, PI-50 and PI−100, respectively. $^1$H-NMR (DMSO-$d_6$, 600 MHz), $\delta_H$ (ppm): 13.48 (s, H, $H_{21}$), 8.12–8.25 (m, 4H, $H_6$), 7.96 (s, 5H, $H_7$ and $H_{16}$), 7.85 (s, 4H, $H_3$), 7.73–7.82 (d, 2H, $H_{17}$), 7.45–7.54 (d, 4H, $H_{12}$) 7.2–7.29 (d, 4H, $H_{13}$), 6.48 and 6.4 (s, 2H, $H_{21}$), 4.56 (d, 2H, $H_{22}$), 4.48 (d, 2H, $H_{24}$), 2.01 (d, 3H, $H_{25}$).

### Preparation of crosslinked polyimide (CPI) aerogel fibers

The PPI, Irgacure2100 and NMP with a weight ratio of 15:0.2:84.8 were mixed by mechanical stirring to obtain the PPI spinning solution. Then, the CPI aerogel fiber was achieved via wet spinning with UV irradiation, solvent exchange and ambient pressure drying. The PPI spinning solution was spun through a steel needle (21 G) into a NMP bath with the irradiation by UV light (Omnicure S1500, UV intensity: 2 W cm$^{-2}$) for gelation. The flow rate of the solution was controlled by a syringe pump (TYD01, Lead Fluid Technology Co., Ltd, China) to maintain at 25–200 mm min$^{-1}$. Subsequently, the CPI gel fibers were subjected to solvent exchange (ethanol, 5 h) and ambient pressure drying (25 °C, 2 h) to obtain CPI aerogel fibers.

### Preparation of crosslinked polyimide/phase change material (CPI/PCM) fabric

The CPI aerogel fabric was dipped in paraffin wax in a vacuum oven at 80 °C. Then a filter paper was repeatedly used to remove the excess paraffin adhering to the fiber surface, and finally the CPI/PCM shape memory fabric was obtained.

### Preparation of graphene oxide (GO) gel via ionic cross-linking

GO gel network was formed by adding Ca$^{2+}$ as ionic cross-linking agent[7]. First, the GO solution (10 mg·mL$^{-1}$) was obtained as previously reported[43,44]. Then the GO solution (10 mL) was added into CaCl$_2$ solution and the mass ratio of CaCl$_2$ to GO was 3:8.

### Preparation of Silica (SiO$_2$) gel via condensation reaction

SiO$_2$ gel was fabricated through condensation reaction[16]. The acetic acid (5 g) was added to 75 g of water and mixed for 30 min. Then CTAB (0.5 g), urea (2.5 g) and methyltrimethoxysilane (5 g) were sequentially added into the above solution. Finally, the mixture was heated to 80 °C to form a gel.

### Preparation of Polyimide (PI) gel via chemical crosslinking

Polyimide gels were synthesized from the dianhydride (BPDA) and the diamines (DMBZ) with crosslinking agent (TAB)[16]. In detail, BPDA (0.606 g, 2.06 mmol) was gradually put into the mixed solution of DMBZ (0.242 g, 2 mmol) and NMP (7.34 mL). After 4 hours of stirring at room temperature, the PAA solution was formed. Subsequently, TAB (13.8 mg, 0.04 mmol) dissolved in 0.92 mL NMP was injected into the above solution and stirred for 5 min. Finally, acetic anhydride (1.556 mL, 16.64 mmol) and pyridine (1.326 mL, 16.64 mmol) were added for imidization. The PI gel was obtained through chemical crosslinking reaction.

### Characterization and measurement

The chemical structures were recorded using a FTIR spectrometer (Nicolet iS50, Thermo Scientific, Germany) and a nuclear magnetic resonance spectrometer (Bruker Avance 600, Bruke, Germany). For NMR tests, the samples were dissolved in deuterated dimethyl sulfoxide (DMSO-$d_6$). Dynamic rheology experiments were performed on a photo-rheometer (MCR 302, Anton Paar, China) with parallel-plate (APCN-LAP-344, 25-mm diameter) geometry and a 365 nm UV lamp (30 mW cm$^{-2}$). The porous structure and cross-section morphology of the CPI aerogel fiber were observed by field-emission scanning electron microscope (FESEM, JSM-7500F, JEOL, Japan). Mechanical properties of aerogel fibers were tested by an electronic universal testing machine (UTM2102, Suns Technology Stock Co., Ltd, China) with a sensor of 50 N and 2 mm min$^{-1}$. The thermal conductivity ($\lambda$) of fabrics was measured with a hot disk thermal analyzer (TPS 2500 S, Hot Disk, Sweden) under heating power of 5 mW and heating time 10 s. In detail, the Kapton probe (type 7281) was placed in the middle of two pieces of fabric (length × width × height: 80 mm × 80 mm × 2 mm), and a certain pressure (500 g) was applied above the fabric, followed by stabilization for 60 min to start the test. The thermal images were captured by a thermal imager (FOTRIC 220 S, FOTRIC, Germany). Phase change temperature of CPI/PCM fiber was tested on a differential scanning calorimeter (TGA/DSC1/1100LF, METTLER, Switzerland). The molecular weight of polymer was recorded using gel permeate chromatography (Agilent PL-GPC50, Agilent, USA). DMF was used as an eluant at flow rate of 1.0 mL min$^{-1}$. In GPC analysis, calibration is based on polymethyl methacrylate standards. A system of three columns was used 2 × PL-gel Mixed-b (300 × 7.5 mm).

The gelation depth of the gel fibers can be described by the Jacob's equation and as Eq. (5).

$$C_d = D_p \ln\left(\frac{E_0}{E_C}\right) \tag{5}$$

where $C_d$ is the cure depth, $E_0$ is the energy density of incident light, $E_c$ is the critical energy density or the minimum energy for the ink to be solidified, and $D_p$ is the PPI solution sensitivity[45].

The double bond conversion ($X$) at time $t$ was calculated by internal standard method as Eq. (6)[32]:

$$X = 1 - \frac{[M]}{[M]_0} = \frac{(A_{1628})_0 - (A_{1628})_t}{(A_{1628})_0} \tag{6}$$

which $(A_{1628})_0$ and $(A_{1628})_t$ is the peak area of the C = C absorption peak in FTIR spectra at irradiation time of 0 and $t$, respectively.

## Data availability

The data supporting the findings of this study are available within the article and its Supplementary Information as well as Source Data. Source data are provided with this paper.

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

## Acknowledgements

This work was supported by the National Natural Science Foundation of China (52373076 and 52073053 to W.F., 52233006 to T.L.), National Key Research and Development Program of China (2022YFA1203600 to W.F.), Young Elite Scientists Sponsorship Program by CAST (2021QNRC001 to W.F.), Shanghai Rising-Star Program (21QA1400300 to W.F.), Innovation Program of Shanghai Municipal Education Commission (2021-01-07-00-03-E00108 to T.L.).

## Author contributions

T.X.: Methodology, Investigation, Writing-Original draft preparation; C.Z. & D.Y.: Investigation, Data curation; X.Z.: Coarse-grained molecular dynamics simulations; F.L., L.Z. & C.Z.: Formal analysis, Data curation; W.F. & T.L.: Funding acquisition, Conceptualization, Methodology Supervision, Writing- Reviewing and Editing.

## Competing interests

The authors declare no competing interests.
