## [Peer review file · Nature Communications]

REVIEWER COMMENTS

Reviewer #1 (Remarks to the Author):

The authors present a novel wet spinning method incorporating a UV-enhanced dynamic gelation strategy for the large-scale preparation of lightweight, high strength crosslinked polyimide aerogel fibers. This novel strategy achieves a fast sol-gel kinetic process as well as the construction of a high-strength gel skeleton, which solves the bottlenecks in the preparation of polyimide aerogel fibers. In addition, the manuscript includes proof-of-concept demonstrations of potential applications for polyimide aerogel fiber, such as intelligent thermally adaptive fabric, to provide ideas for the development of new thermal management fabric. I recommend the manuscript for the publication after addressing the following minor issues.

1. Why is it not feasible to produce polyimide aerogel fibers using conventional wet spinning methods? Furthermore, what is the microstructure of the fiber once the UV-enhanced dynamic gelation strategy is eliminated?

2. In general, the molecular weight of photosensitive resins plays a crucial role in determining fluidity, gelation rate and gelation depth. Therefore, please supplement the molecular weight of PPI.

3. Crosslinked polyimide aerogel fibers with integrated phase change materials are used as intelligent thermally adaptive fabric. It is well known that phase change materials are susceptible to leakage during solid-liquid conversion. Therefore, the authors need to consider the cyclic stability of CPI/PCM.

4. Furthermore, I would like to inquire about the potential impact of impregnating crosslinked polyimide aerogel fibers with phase change materials on their mechanical properties. Could you kindly provide your insights on this?

5. For some characterizations and measurement, authors should provide details of the test procedure. For example, the test of thermal conductivity.

Reviewer #2 (Remarks to the Author):

An interesting preparation and application of crosslinked polyimide fibers has been described. This study has made significant advances over previous studies and the results are interesting and relevant.

General comments:

-The structures under investigation contain -CF₃ groups. The general trend is to reduce and remove the fluorine from materials; several regulatory restrictions are under discussion and some bans have been recently enforced (e.g., F-compounds have been recently banned in ski waxes in EU). Why have you chosen CF₃ containing polymers? Is it inevitable in this context? What about the sustainability and end-of life aspects of your materials?

-The mechanical properties of fibers have been thoroughly characterized, however, the polymer synthesis and especially (polymer) structural characterization should be improved. For example:
* the synthesis procedure for PI (lines 427-435) is, in general, understandable, but how is the synthesis outcome verified? How is the completion of imidization evaluated? What was the conversion/yield etc.? What is the PI solubility? How is the structure confirmed and what is the molecular weight of the polymer obtained? MW and polydispersity data could be crucial for reproducibility and to achieve fibers with desired characteristics. SEC can also be run in NMP. There

are some highly compressed ^1H NMR-s in ESI, but no data on NMR conditions (e.g. solvent) is listed. I recommend including full size NMR-s to ESI for better evaluation. The disappearance of certain peaks in FTIR can only indicate the presence or disappearance of certain bonds, but not e.g., the length of polymer.

* PPI series. Can the authors comment the real grafting ratio? Is it comparable with the targeted ratio? What is the solubility of PPI and is it stable upon storage, can self-polymerization (crosslinking) occur?

Some additional minor issues:

Line 51, should be Kevlar

Line 120, should be Steglich esterification

Line 417, should be dianhydride

Line 422, dicyclohexyl carbodiimide is much more commonly used name for DCC

Line 431, ..diamines were completely...

-Figure 3di UV-enhanced gelation. Check the number of carbons on the product side.

-ESI Supplementary Figure 5. Check the number of carbons in crosslinked structural fragments in red.

To summarize, I recommend considering publishing this manuscript after the characterization of polymers and polymerization process has been properly described. Such proper description would ensure the reproducibility of results and might open up for new application areas.

Point-to-point response to reviewers

Reviewer #1 (Remarks to the Author):

The authors present a novel wet spinning method incorporating a UV-enhanced dynamic gelation strategy for the large-scale preparation of lightweight, high strength crosslinked polyimide aerogel fibers. This novel strategy achieves a fast sol-gel kinetic process as well as the construction of a high-strength gel skeleton, which solves the bottlenecks in the preparation of polyimide aerogel fibers. In addition, the manuscript includes proof-of-concept demonstrations of potential applications for polyimide aerogel fiber, such as intelligent thermally adaptive fabric, to provide ideas for the development of new thermal management fabric. I recommend the manuscript for the publication after addressing the following minor issues.

Author Reply: We are thankful that the reviewer has a positive opinion about our work.

We have carefully addressed this reviewer's comments to further improve the quality of this paper.

1. Why is it not feasible to produce polyimide aerogel fibers using conventional wet spinning methods? Furthermore, what is the microstructure of the fiber once the UV-enhanced dynamic gelation strategy is eliminated?

Author Reply: We are grateful for the questions raised by the reviewer. Aerogel fibers are produced through the extrusion of the polymer solution via a spinneret followed by a sol-gel transition to form gel fibers, with subsequent solvent exchange and specific drying. In this process, the sol-gel transition process is essential for the construction of unique aerogel microstructures. Conventional wet spinning is a fiber spinning process used to produce synthetic fibers. It involves the extrusion of a polymer solution through a spinneret into a coagulation bath where the fibers solidify and form. The coagulation bath contains a non-solvent for the polymer, causing the solvent in the spinneret dope to rapidly diffuse out of the fiber. Conventional wet spinning is based on the principle of phase separation without the sol-gel process, where molecular chains exhibit strong motility. As the solvent is extracted, solid fibers would be formed. As a result of the inability to form a three-dimensional gel backbone in conventional wet spinning, the final fibers failed to exhibit the typical three-dimensional porous aerogel structure. Therefore, the production of polyimide aerogel fibers using conventional wet spinning methods is not feasible.

To demonstrate the infeasibility of conventional wet spinning, we removed the UV irradiation during spinning process and prepared non-crosslinked PPI-100 fibers by squeezing the PPI-100 spinning solution directly into the coagulation bath (ethanol). As shown in Fig. R1, the polymer chains started to aggregate together under the effect of non-solvent, resulting in the PPI-100 fibers showing a typical finger-like pore structure. In contrast, crosslinked polyimide aerogel (CPI) fibers prepared via UV-enhanced dynamic gelation strategy show highly porous nanostructures formed by 3D

interconnected nanofiber networks, demonstrating typical aerogel morphology (Fig. 1d in revised manuscript).

Fig. R1 Cross-sectional SEM images of PPI-100 fiber at a) low and b) high magnifications. The PPI-100 fibers were prepared by squeezing the PPI-100 spinning solution into coagulation bath (ethanol) and leaving it for 30 min, followed by successive ethanol washes and ambient pressure drying.

Added text, page 17, top: “However, after removal of the UV dynamic gelation process, the PPI polymer chains started to aggregate together in the presence of non-solvents, resulting in the PPI fibers displaying a typical finger-like pore structure (Supplementary Fig. 17).”

2. In general, the molecular weight of photosensitive resins plays a crucial role in determining fluidity, gelation rate and gelation depth. Therefore, please supplement the molecular weight of PPI.

Author Reply: Thank you for your valuable advice. According to the suggestions from the reviewer, we supplemented the molecular weight of PPI in Supplementary Fig. 5.

In addition, the corresponding discussion is supplemented in line 1-2 on page 8.

Molecular weights were determined by gel permeate chromatography (GPC, Agilent PL-GPC50, Agilent, USA). *N,N*-dimethylformamide (DMF) was used as an eluant at flow rate of 1.0 mL min⁻¹. The molecular weight distribution of PPI-100 is shown in Fig. R2. The PPI-100 showed high relative molecular weight ($M_w \sim 31205 \text{ g mol}^{-1}$) and polydispersity index (PDI \sim 1.73), which is applicable to typical photo-crosslinking processes (Advanced Materials, 2017, 29, 1701240). Importantly, the mechanical properties of the aerogel fiber are strongly related to the molecular weight of the polymer. The high molecular weight of PPI ensures the mechanical properties of the aerogel fibers, and also enables successful UV dynamic gelation.

Fig. R2 Molecular weight distribution curve of PPI-100.

Added text, page 7, bottom: “The molecular weight of PPI is measured as 31205 g mol⁻¹, which is applicable to typical photo-crosslinking processes²⁹ (Supplementary Fig. 5).”

3. Crosslinked polyimide aerogel fibers with integrated phase change materials are used as intelligent thermally adaptive fabric. It is well known that phase change materials

are susceptible to leakage during solid-liquid conversion. Therefore, the authors need to consider the cyclic stability of CPI/PCM.

Author Reply: We sincerely appreciate the valuable suggestions provided by the reviewer. To demonstrate the cyclic stability of CPI/PCM fabrics, heating-cooling cycle tests were performed on CPI/PCM. As shown in Fig. R3a, the mass retention ratio of the CPI/PCM fabric after 50 heating-cooling cycles was 96.7%. Furthermore, thermal properties of CPI/PCM fabric after different heating and cooling cycle were investigated (Fig. R3b). The DSC curves after 50 heating-cooling cycles are almost coincident with that of the original one, reflecting the cyclic stability of PI/PCM fabric. This is mainly contributed to the fact that the CPI aerogel fibers with a high specific surface area and abundant mesopores would have strong capillary force to confine phase-change materials without leakage. According to the suggestions from the reviewer, we supplemented the cyclic stability of CPI/PCM fabric in Supplementary Fig. 31, and the corresponding discussion is supplemented in line 20-21 on page 21.

Fig. R3 (a) The mass retention ratio of CPI/PCM fabric after different heating and cooling cycle. (b) Differential scanning calorimetric (DSC) curves of CPI/PCM fabric after different heating and cooling cycle.

Added text, page 21, bottom: “This is mainly attributed to the stable thermal behavior of CPI/PCM fabrics in heating-cooling cycles (Supplementary Fig. 31).”

4. Furthermore, I would like to inquire about the potential impact of impregnating crosslinked polyimide aerogel fibers with phase change materials on their mechanical properties. Could you kindly provide your insights on this?

Author Reply: We are grateful for the questions raised by the reviewer. The integration of PCM can enhance the mechanical properties of CPI aerogel fiber. The fracture strength (45.8 MPa) of CPI/PCM fiber is higher than CPI aerogel fiber (22 MPa) and the Young’s modulus of CPI/PCM fiber (1000 MPa) is five times higher than CPI aerogel fiber (200 MPa). This mainly attributed to the fact that the PCM impregnated in the porous structure can help to transfer the load and facilitates the dispersion and dissipation of stress in the matrix. We added Supplementary Fig. 25f to show the mechanical properties of the CPI/PCM fiber.

Fig. R4 Tensile stress-strain curve of CPI fiber and CPI/PCM fiber.

Added text, page 20, bottom: “Besides, PCM filling in the porous structure of CPI aerogel fibers can improve the mechanical strength of PI/PCM composite fibers (Supplementary Fig. 25f).”

5. For some characterizations and measurement, authors should provide details of the test procedure. For example, the test of thermal conductivity.

Author Reply: Thank you for pointing this out. The corresponding details is presented on page 26 and page 27 of the revised manuscript. In this work, the thermal conductivity of the fabric was measured by the Hot Disk transient planar heat source method under a film test mode. In detail, the Kapton probe (type 7281) was placed in the middle of two pieces of fabric (length \times width \times height: 80 mm \times 80 mm \times 2 mm), and the thermal conductivity of the fabric was tested at a heating power of 5 mW and heating time of 10 s. A certain pressure (500 g) was applied above the fabric during the test. Importantly, it needs to be stabilized for 60 min before testing to ensure the accuracy of the test. The NMR of the polymers were characterized by nuclear magnetic resonance spectrometer (Bruker Avance 600, Bruker, Germany) by dissolving the samples in deuterated dimethyl sulfoxide (DMSO). The molecular weight of PPI was recorded using gel permeate chromatography (Agilent PL-GPC50, Agilent, USA). DMF was used as an eluant at flow rate of 1.0 mL min⁻¹.

Added text, page 26, bottom: “For NMR tests, the samples were dissolved in deuterated dimethyl sulfoxide (DMSO-*d*₆).”

Added text, page 27, top: “In detail, the Kapton probe (type 7281) was placed in the middle of two pieces of fabric (length × width × height: 80 mm×80 mm×2 mm), and a certain pressure (500 g) was applied above the fabric, followed by stabilization for 60 min to start the test.”

Added text, page 27, bottom: “The molecular weight of polymer was recorded using gel permeate chromatography (Agilent PL-GPC50, Agilent, USA). DMF was used as an eluant at flow rate of 1.0 mL min⁻¹.”

Reviewer #2 (Remarks to the Author):

An interesting preparation and application of crosslinked polyimide fibers has been described. This study has made significant advances over previous studies and the results are interesting and relevant.

Author Reply: We are thankful that the reviewer has a positive opinion about our work. We have carefully addressed this reviewer’s comments to further improve the quality of this paper.

General comments:

-The structures under investigation contain $-CF_3$ groups. The general trend is to reduce and remove the fluorine from materials; several regulatory restrictions are under discussion and some bans have been recently enforced (e.g., F-compounds have been recently banned in ski waxes in EU). Why have you chosen CF_3 containing polymers? Is it inevitable in this context? What about the sustainability and end-of life aspects of your materials?

Author Reply: Thank you for pointing this out. It's true that there has been a growing trend in recent years to reduce and eliminate the use of fluorinated compounds, due to concerns about their environmental and health impacts. We will also reduce the use of such polymers in our future work.

In this work, the presence of $-CF_3$ groups is necessary. Firstly, the poor solubility of conventional polyimides (PI) has been an important factor limiting the processing of PI fibers. In detail, insoluble PI are normally processed via the sol-gel transition of soluble poly(amic acid) (PAA) intermediate followed by thermal or chemical imidization. This two-step reaction involves a complex and long-time sol-gel process of PAA, which cannot match with the short-time and continuous spinning process, giving rise to the difficulty in preparing PI aerogel fibers. The introduction of $-CF_3$ groups into the main chain of PI can effectively increase the free volume between molecular chains, reduce the regularity of molecular chain segments, and reduce the π - π interactions between benzene rings, resulting in good solubility and processability. In addition, the PI with abundant $-CF_3$ groups exhibits hydrophobic surface and low capillary pressure, which

can further avoid structural collapse during ambient pressure drying. In summary, the -CF₃ groups confer excellent solubility to the PI and reduces capillary pressure in ambient pressure drying, which is essential for the CPI aerogel fibers.

For the sustainability and end-of life aspects of PI aerogel fiber, polyimides are known for their durability and longevity. Their resistance to heat, chemicals, and wear can extend the lifespan of products in which they are used. This can reduce the need for frequent replacements and, in turn, reduce resource consumption and waste generation. Furthermore, there are emerging technologies and methods for recycling PI materials, such as chemical recycling and pyrolysis. These methods break down polyimides into their constituent monomers, which can then be used to produce new polyimide materials.

-The mechanical properties of fibers have been thoroughly characterized, however, the polymer synthesis and especially (polymer) structural characterization should be improved. For example:

* the synthesis procedure for PI (lines 427-435) is, in general, understandable, but how is the synthesis outcome verified? How is the completion of imidization evaluated? What was the conversion/yield etc.? What is the PI solubility? How is the structure confirmed and what is the molecular weight of the polymer obtained? MW and polydispersity data could be crucial for reproducibility and to achieve fibers with desired characteristics. SEC can also be run in NMP. There are some highly compressed ¹H NMR-s in ESI, but no data on NMR conditions (e.g. solvent) is listed. I recommend

including full size NMR-s to ESI for better evaluation. The disappearance of certain peaks in FTIR can only indicate the presence or disappearance of certain bonds, but not e.g., the length of polymer.

Author Reply: Thank you for pointing this out. According to the suggestions from the reviewer, we supplemented the structural characterization of PI in Supplementary Fig. 2 and corresponding discussion on page 6 and page 7 in the revised manuscript.

To demonstrate the smooth formation of polyimide (PI) from poly(amic acid) (PAA) at elevated temperatures, we characterized the differences in the functional groups of PAA and PI using FTIR. The polyimide showed characteristic imide absorption peaks near 1778 cm^{-1} and 1720 cm^{-1} corresponding to the imide ring of the polyimide. The disappearance of the absorption peak of amide bond (-CONH-) at 1661 cm^{-1} and 1546 cm^{-1} and appearance of new peaks (-C-N-, 1355 cm^{-1}) in the FTIR spectrum of PI indicates that the poly(amic acid) (PAA) is completely imidized during the heating process (Fig. R5b). Moreover, the signals for amides ($\delta_{\text{H}}=10.5\text{-}11.0\text{ ppm}$) was not detected in ^1H NMR, indicating that the imidization and dehydrocyclization reactions of poly(amic acid) proceeded at a high efficiency (Fig. R5c) (Adv. Energy Mater. 2019, 9, 36, 1901987).

Structure of the PI was determined by ^1H NMR, ^{13}C NMR, ^{19}F NMR and FT-IR spectra. The FT-IR spectra of PI showed two peaks assigned to the carbonyl group, 1355 cm^{-1} (C-N stretching) and 1784 cm^{-1} (C=O asymmetric stretching), which represented the characteristic peaks of the polyimide. The absorption peak near 1145

cm^{-1} was assigned to the C-F bond, while the strong peaks appearing at 1100 cm^{-1} was assigned to the ether bond vibration peaks. The peaks at 1498 cm^{-1} and 1451 cm^{-1} were assigned to the C-C stretching of the aromatic rings (Macromolecules 2021, 54, 22, 10271–10278). The ^1H NMR spectrum of PI showed the main chain proton signals for aromatics at 7.2–8.25 ppm (Fig. R5c). Characteristic signals of the carboxyl group (166.6 ppm), the imide ring (166.3 and 166.1 ppm), the C-F and the benzene ring (140–120 ppm) are clearly observed in the ^{13}C NMR pattern. And in ^{19}F NMR spectrum, the characteristic signal of $-\text{CF}_3$ appeared at -63.63 ppm (ACS Appl. Mater. Interfaces 2016, 8, 39, 26352–26358). Therefore, the successful synthesis of the expected PIs was clearly indicated by the NMR and FTIR spectra, and the yield of PI is as high as 98.1%.

Fig. R5 (a) Chemical structure of PI. (b) FTIR spectra of PI and PAA. (c) ^1H NMR

spectrum of PI. (d) ^{13}C NMR spectrum of PI. (e) ^{19}F NMR spectrum of PI.

Added text, page 6, bottom and page 7, top: “The complete conversion of PAA to the PI was also confirmed by the disappearance of the amide (-NHCO-) at 10.5-11.0 ppm in the ^1H NMR spectrum of PI²⁴ (Supplementary Fig. 2c). The chemical structure of PI was further confirmed by ^1H NMR, ^{13}C NMR, ^{19}F NMR, and FTIR spectra (Supplementary Fig. 2).”

The solubility was determined using 10 mg of PI in 1 mL of solvent. The introduction of the $-\text{CF}_3$ group in the PI main chain can effectively increase the free volume between the molecular chains, reduce the regularity of the molecular chain segments, and reduce the π - π interactions between the benzene rings, thus obtaining good solubility and processability. Therefore, the PI dissolves in strong polar solvents at room temperature, such as *N*-methylpyrrolidone (NMP), *N,N*-dimethylformamide (DMF) and *N,N*-dimethylacetamide (DMAc) and so on (Table R1).

Table R1 Solubility of PI.

Sample	Solvent							
	NMP	DMAc	DMF	DMSO	Acetone	CH_2Cl_2	CHCl_3	THF
PI	++	++	++	++	++	+-	+-	++

++: soluble at room temperature. +-: partial swelling.

Added text, page 7, top: “The synthesized PI is soluble in organic solvents such as *N*-methylpyrrolidone (NMP), *N,N*-dimethylformamide (DMF) and *N,N*-dimethylacetamide (DMAc) and so on (Supplementary Table 1), since the introduction of the -CF₃ group in the PI main chain can effectively increase the free volume between the molecular chains, reduce the regularity of the molecular chain segments, and reduce the π - π interactions between the benzene rings, resulting in the good solubility.”

We supplemented the molecular weight of PI in Supplementary Fig. 3. In addition, the corresponding discussion is supplemented in line 7-9 on page 7. Molecular weights were determined by gel permeate chromatography (GPC, Agilent PL-GPC50, Agilent, USA). DMF was used as an eluant at flow rate of 1.0 mL min⁻¹. The molecular weight distribution of PI is shown in Fig. R6. The PI showed high relative molecular weight ($M_w \sim 30273$ g mol⁻¹) and low polydispersity index (PDI ~ 1.63).

Fig. R6 Molecular weight distribution curve of PI.

Finally, the details of the NMR tests were added and the full-size NMR spectrum were adjusted in Supplementary Fig. 4 according to the reviewers' suggestions.

Added text, page 7, top: “The resulting organo-soluble PI showed high molecular weight ($M_w \sim 30273 \text{ g mol}^{-1}$) and narrow polydispersity index (PDI \sim 1.63) (Supplementary Fig. 3).”

* PPI series. Can the authors comment the real grafting ratio? Is it comparable with the targeted ratio? What is the solubility of PPI and is it stable upon storage, can self-polymerization (crosslinking) occur?

Author Reply: Thank you for your constructive advice. According to the suggestions from the reviewer, we calculated the practical grafting ratio (GR) of HEMA on PPI-x based on ^1H NMR, supplemented in Supplementary Fig. 4d. In addition, the corresponding discussion is supplemented in line 20-22 on page 7. When the theoretical grafting ratio of HEMA was 25%, 50% and 100%, the practical grafting ratio calculated by ^1H NMR was 24.8%, 47.9% and 76.0%, respectively (Fig. R7). The practical grafting ratio is comparable with the target grafting ratio.

Fig. R7 Practical grafting ratio (GR) of PPI with different HEMA grafting ratios.

**Added text, page 7, bottom:** "The practical grafting ratio calculated by $^1H$ NMR are 24.8%, 47.9% and 76.0% for PPI-25, PPI-50 and PPI-100, respectively, which are comparable with the theoretical grafting ratio (Supplementary Fig. 4d)."

The solubility was determined using 10 mg of PPI in 1 mL of solvent, supplemented in Supplementary Table 3. The corresponding discussion is supplemented in line 4-6 on page 8. PPI exhibited exceptional solubility in common solvents like *N*-methylpyrrolidone (NMP), *N,N*-dimethylacetamide (DMAc), *N,N*-dimethylformamide (DMF), acetone, dichloromethane ($CH_2Cl_2$), trichloromethane ($CHCl_3$) and tetrahydrofuran (THF) at room temperature (Table R2). The good solubility of the PPI is mainly attributed to the introduction of trifluoromethyl groups in the main chain and the presence of HEMA group, which can effectively increase the free volume between molecular chains and reduce the regularity of molecular segments to reduce the π-π interactions between benzene rings, resulting in good solubility. The good solubility of PPI in common solvents allows for the formation of spinning solution for wet spinning and improve processability.

**Table R2** Solubility of PPI-100.

| **Sample** | **Solvent** | | | | | | | |
|---|---|---|---|---|---|---|---|---|
| | **NMP** | **DMAc** | **DMF** | **DMSO** | **Acetone** | **$CH_2Cl_2$** | **$CHCl_3$** | **THF** |
| **PPI** | ++ | ++ | ++ | ++ | ++ | ++ | ++ | ++ |

++: soluble at room temperature.

According to the suggestions from the reviewer, the stability of the PPI solution during storage is supplemented in Supplementary Fig. 6. In addition, the corresponding discussion is supplemented in line 4-6 on page 8. The PPI solution without photoinitiator (Irgacure 2100) gelled after 30 days in natural light (sample a). The PPI solution with photoinitiator fast gelled after 1 day in natural light (sample b). However, it is worth noting that the PPI solution kept in dark can be stored stably for 30 days when the bottle is wrapped in aluminum foil (sample c, d). This storage stability facilitates spinning processing.

Fig. R8 Storage stability experiments of PPI spinning solution. (a) PPI solution without photoinitiator (Irgacure2100). (b) PPI solution with photoinitiator (Irgacure2100). (c) PPI solution without photoinitiator (Irgacure2100) stored away from light. (d) PPI solution with photoinitiator (Irgacure2100) stored away from light.

Added text, page 8, top: “The PPI exhibits excellent solubility and stability in common

solvents, which ensures the formation of spinning solution and improves processability (Supplementary Fig. 6 & Supplementary Table 3).”

Some additional minor issues:

Line 51, should be Kevlar

Line 120, should be Steglich esterification

Line 417, should be dianhydride

Line 422, dicyclohexyl carbodiimide is much more commonly used name for DCC

Line 431, ..diamines were completely...

-Figure 3di UV-enhanced gelation. Check the number of carbons on the product side.

-ESI Supplementary Figure 5. Check the number of carbons in crosslinked structural fragments in red.

Author Reply: Thank you for pointing this out. As suggested by the reviewer, we have checked the manuscript carefully, corrected writing mistakes and polished the manuscript. In addition, according to the suggestions from the reviewer, we have revised the Figure 3d and Supplementary Fig. 8 in the revised manuscript.

To summarize, I recommend considering publishing this manuscript after the characterization of polymers and polymerization process has been properly described. Such proper description would ensure the reproducibility of results and might open up for new application areas.

REVIEWERS' COMMENTS

Reviewer #1 (Remarks to the Author):

I am satisfied with changes authors made according to my suggestions and thus recommend it publish in its current state.

Reviewer #2 (Remarks to the Author):

The authors have done a thorough job with this revision and improved the overall quality of the manuscript significantly. Polymer characterization data has been added and the relevant discussion sections updated. MW of polymers has been added. My only minor comment is that the authors should also add which reference polymers were used to calibrate the GPC system and which columns were used.

Point-to-point response to reviewers

Reviewer #1 (Remarks to the Author):

I am satisfied with changes authors made according to my suggestions and thus recommend it publish in its current state.

Author Reply: We sincerely thank the reviewer for recommending accepting our manuscript.

Reviewer #2 (Remarks to the Author):

The authors have done a thorough job with this revision and improved the overall quality of the manuscript significantly. Polymer characterization data has been added and the relevant discussion sections updated. M_w of polymers has been added. My only minor comment is that the authors should also add which reference polymers were used to calibrate the GPC system and which columns were used.

Author Reply: We sincerely appreciate the valuable suggestions provided by the reviewer. Based on your feedback, we have added the corresponding discussion in the revised manuscript on page 28. In GPC analysis, calibration is based on polymethyl methacrylate standards. A system of three columns was used $2 \times$ PL-gel Mixed-b (300

× 7.5 mm).

Added text, page 28, top: “In GPC analysis, calibration is based on polymethyl methacrylate standards. A system of three columns was used 2 × PL-gel Mixed-b (300 × 7.5 mm).”